**Data Availability Statement:** The data underlying the Bangladesh birth cohort used for the primary analysis are available on the Harvard Dataverse:

# Pregnancy pesticide exposure and child development in low- and middle-income countries: A prospective analysis of a birth cohort in rural Bangladesh and meta-analysis

Lilia Bliznashka[1,2]*, Aditi Roy[3], David C. Christiani[4], Antonia M. Calafat[5], Maria Ospina[5], Nancy Diao[4], Maitreyi Mazumdar[4,6], Lindsay M. Jaacks[2,3]

1 Nutrition, Diets, and Health Unit, International Food Policy Research Institute, Washington, DC, United States of America, 2 Global Academy of Agriculture and Food Systems, University of Edinburgh, Scotland, United Kingdom, 3 Center for Environmental Health, Public Health Foundation of India, New Delhi, India, 4 Department of Environmental Health, Harvard T.H. Chan School of Public Health, Boston, MA, United States of America, 5 Division of Laboratory Sciences, Centers for Disease Control and Prevention, Atlanta, GA, United States of America, 6 Department of Neurology, Boston Children's Hospital, Boston, MA, United States of America

* l.bliznashka@cgiar.org

## Abstract

### Background

Despite considerable evidence on a negative association between pregnancy pesticide exposure and child development in high-income countries, evidence from low- and middle-income countries (LMICs) is limited. Therefore, we assessed associations between pregnancy pesticide exposure and child development in rural Bangladesh and summarised existing literature in a systematic review and meta-analysis.

### Methods

We used data from 284 mother-child pairs participating in a birth cohort established in 2008. Eight urinary pesticide biomarkers were quantified in early pregnancy (mean gestational age 11.6±2.9 weeks) as an index of pesticide exposure. The Bayley Scales of Infant and Toddler Development, Third Edition were administered at 20–40 months of age. Associations between creatinine-adjusted urinary pesticide biomarker concentrations and child development scores were estimated using multivariable generalised linear models. We searched ten databases up to November 2021 to identify prospective studies on pregnancy pesticide exposure and child development conducted in LMICs. We used a random-effects model to pool similar studies, including our original analysis. The systematic review was pre-registered with PROSPERO: CRD42021292919.

### Results

In the Bangladesh cohort, pregnancy 2-isopropyl-4-methyl-6-hydroxypyrimidine (IMPY) concentrations were inversely associated with motor development (-0.66 points [95% CI -1.23, -0.09]). Pregnancy 3,5,6-trichloro-2-pyridinol (TCPY) concentrations were inversely

https://doi.org/10.7910/DVN/DIZVBV. All data used for the systematic review and meta-analysis are included in the paper and appendix.

**Funding:** Funding for the prospective study was provided by the Burke Global Health Fellowship program at the Harvard Global Health Institute and the National Institutes of Health (R01-ES015533, P30-ES00002, P42-ES016454). LB and LMJ were supported by the Medical Research Council/UK Research and Innovation (Grant Ref: MR/T044527/1). AR was supported by a DBT/Wellcome Trust India Alliance Fellowship (grant number IA/CPHE/20/1/505272). For the purpose of open access, the author has applied a Creative Commons Attribution (CC BY) licence to any Author Accepted Manuscript version arising from this submission.

**Competing interests:** The authors have declared that no competing interests exist.

associated with cognitive development, but the association was small: -0.02 points (-0.04, 0.01). We observed no associations between 4-nitrophenol and 3-phenoxybenzoic acid (3-PBA) concentrations and child development. The systematic review included 13 studies from four LMICs. After pooling our results with one other study, we found consistent evidence that pregnancy 3-PBA concentrations were not associated with cognitive, language, or motor development.

## Conclusion

Evidence suggests that pregnancy exposure to some organophosphate pesticides is negatively associated with child development. Interventions to reduce in-utero pesticide exposure in LMICs may help protect child development.

## 1. Introduction

Pesticide use in Asia has increased dramatically over the past 40 years [1, 2]. In Bangladesh, 80% of farmers use pesticides at least once per crop season [3]. Overuse is widespread particularly in vegetable farming, including the use of banned pesticides like dichlorodiphenyltrichloroethane [4, 5]. In some parts of the country, pesticide residues, including organophosphates and pyrethroids, are frequently detected in vegetables and fruit sold in markets [6–9] High concentrations of organophosphate, pyrethroid, and carbamate residues are also frequently found in water and soil [10–12].

Widespread exposure to pesticides results in numerous carcinogenic, reproductive, immunological, neurological, and other adverse health effects in adults [13, 14]. Compared to adults, children are especially vulnerable to the harmful effects of pesticides because of their increased exposure relative to their body weight [15, 16] and dynamic developmental physiology [17, 18]. Behavioural factors (e.g., crawling) also play a role [15]. Maternal pesticide exposure during pregnancy is of particular concern because of transplacental transfer [19, 20] and documented effects of in-utero pesticide exposure on brain development through inhibition of acetylcholinesterase (AChE) activity [21], and cortical thinning [22]. Animal studies suggest that pesticide exposure in early life may result in long-term irreversible changes in the nervous system [17, 23]. Long-latency delayed neurotoxicity, where neurotoxicity presents itself years after exposure has ceased, may also occur given the brain's plasticity in early life, potential neuronal compensation, and potential of historic neurotoxicant exposure to accelerate the normal decline in neurotransmitters and repair mechanisms which occurs with age [24]. Thus, consequences of functional damage resulting from early life neurotoxins may emerge later in life [15].

Extensive evidence has linked pregnancy exposure to organophosphates with cognitive, motor, and behavioural delays in infancy, childhood, and adolescence [25–28]. However, most of this evidence comes from studies conducted in high-income countries (HICs). One recent review concluded that considerable evidence linked prenatal exposure to organophosphates to child neurodevelopment disorders based on 50 articles, 15 from LMICs [25]. Importantly, that systematic review included only two countries in Asia: China (n = 5) and Thailand (n = 3) [25]. Another recent review concluded that prenatal occupational exposure to pesticides was associated with delays in motor and cognitive development based on 23 studies, four from LMICs (three from Ecuador and one from China) [28]. No studies from Bangladesh or other South Asian countries were included in either review. Yet another recent review concluded

that there was sufficient evidence of an adverse association between prenatal pyrethroid exposure and child neurodevelopment, based on 17 studies only four of which were in LMICs: China (n = 2), Mexico (n = 1), and South Africa (n = 1) [29].

Although prior reviews included LMICs, none disaggregated results by LMICs and HICs, an important distinction given that approved and commonly used pesticides vary between LMICs and HICs [30]. Moreover, these reviews included studies from only four LMICs (China, Ecuador, South Africa, and Thailand), highlighting the scarcity of evidence. In addition, recent reviews have focused more on neurodevelopment disorders [25, 28], even though poor child development without resulting in neurodevelopment disorders has been widely linked with long-term loss of human capital [31]. Lastly, recent reviews synthesised literature on both prenatal and postnatal pesticide exposure. A narrower focus on pregnancy as a particularly sensitive window can help improve our understanding of the relationship between pesticide exposure and child development. Given these limitations of the existing literature, our objective was to strengthen the evidence on pregnancy pesticide exposure and child development in LMICs by conducting a primary analysis using data from rural Bangladesh and a systematic review and meta-analysis to summarise existing evidence from LMICs.

## 2. Materials and methods

### 2.1 Bangladesh cohort

We used data from a prospective birth cohort established in 2008 in rural Bangladesh to assess the effect of early life exposure to heavy metals on child health [32]. Between 2008–2011, 1,613 women meeting the following criteria were enrolled: $\geq$18 years of age, singleton pregnancy <16 weeks' gestation, primary drinking water source was a well, no plans to move before delivery, and planned to deliver at a health centre or at home [32]. At enrolment (mean gestational age 11.6±2.9 weeks), urine samples were collected from a sub-sample of 289 women [33]. Women and their children were followed up when the child was 20–40 months old.

Information on urine sample collection and storage has been previously published [33]. Briefly, trained healthcare workers collected women's urine samples at a health clinic and immediately stored them at -20˚C. Frozen urine samples were shipped on dry ice to Taipei Medical University, Taiwan, where they were stored at -80˚C. Creatinine was measured using a colorimetric assay on a Roche Modular P800 instrument (Roche Inc., Mannheim, Germany) by Taipei Medical University. Remaining urine samples were shipped to the Harvard T.H. Chan School of Public Health on dry ice and stored at -80˚C until they were shipped frozen overnight to the Centers for Disease Control and Prevention (CDC) in December 2017 and August 2018 for analysis. CDC methodologies for the quantification of urinary pesticide biomarkers have been previously described [33, 34]. We measured specific pesticide biomarkers which are more stable over time and under temperature gradients [35, 36]. The pesticide biomarkers measured were: 2,4-dichlorophenoxyacetic acid (2,4-D); 3,5,6-trichloro-2-pyridinol (TCPY); 4-nitrophenol; malathion dicarboxylic acid (MDA); 2-isopropyl-4-methyl-6-hydroxypyrimidine (IMPY); 4-fluoro-3-phenoxybenzoic acid (4-F-3-PBA); 3-phenoxybenzoic acid (3-PBA); and trans-3-(2,2-dichlorovinyl)-2,2-dimethylcyclopropane carboxylic acid (trans-DCCA) [34]. Concentrations below the limit of detection (LOD, 0.3 µg/L for 2,4-D, 0.1 µg/L for TCPY, 1.0 µg/L for MDA, 0.6 µg/L for trans-DCCA, and 0.2 µg/L for all other biomarkers) were assigned a value equal to LOD divided by $\sqrt{2}$ [37]. Four pesticide biomarkers detected in <10% of samples (2,4-D, MDA, trans-DCCA, and 4-F-3-PBA) were excluded from the analyses. We used creatinine-adjusted urinary concentrations (µg/g creatinine) for all analyses. As a sensitivity analysis, concentrations of pesticide biomarkers detected in 10–60% of samples

were treated as binary variables (detected vs. non-detected); concentrations of pesticide biomarkers detected in ≥60% of samples were categorised into terciles.

Child development at 20–40 months of age was assessed using a translated and culturally-adapted version of the Bayley Scales of Infant and Toddler Development, Third Edition (BSID-III) [38]. BSID-III was administered at health clinics by trained staff. We calculated cognitive, language, and motor composite scores (mean = 100, SD = 15) by converting raw scores to scaled composite scores [38].

The analytic sample included 284 mother-child pairs with data on maternal pregnancy pesticide biomarkers and child development at 20–40 months of age. We used *t*-tests to test for differences between mothers in our sample and the rest of the enrolment sample, and between children in our sample and the rest of the children assessed at 20–40 months of age. Differences were considered significant at $p<0.05$. To assess the associations between pregnancy pesticide biomarkers and child development, we fit linear models and calculated unadjusted and adjusted mean differences (MD). Adjusted estimates controlled for an *a priori* set of confounders, selected using a previously published Direct Acyclic Graph [33]: child age at assessment, child sex, maternal age at enrolment, maternal education at enrolment, and maternal energy, vegetable, and fruit intake (assessed using a semi-quantitative food frequency questionnaire [39] administered at 28 weeks' gestation), husband's occupation at enrolment (agricultural work vs. not), and household income at enrolment. There were no missing data in the analytic sample. We explored whether the adjusted associations differed across child sex, maternal education, household income, and husband's occupation. We considered interactions significant at $p<0.10$. All analyses were conducted in Stata 17 [40].

The Bangladesh study was approved by the Institutional Review Boards of the Harvard T. H. Chan School of Public Health (protocol number IRB17-1036) and the Dhaka Community Hospital (protocol number not available). Written informed consent was obtained from all women. The involvement of the Centers for Disease Control and Prevention (CDC) laboratory did not constitute human subjects' research.

## 2.2 Systematic review and meta-analysis

We searched PubMed, Cochrane Library, Embase, Scopus, LILACS, Web of Science, CAB abstracts, Global Health (CABI), Global Index Medicus, and SciELO from inception through November 2021 with no language restriction. LB, AR, and LMJ developed the search strategy (**S1 Table**), informed by prior reviews [25–27] and through consultations with a research librarian. We included peer-reviewed articles meeting the following inclusion criteria: conducted in a LMIC; assessed children <18 years; evaluated self-reported exposure to pesticides or measured pesticide biomarkers in pregnancy (at a single or multiple time points); measured at least one child development outcome; and was a prospective study design. We excluded animal studies, case-control studies, cross-sectional studies, simulation studies, case reports, case studies, opinions, editorials, commentaries, letters, conference abstracts, ecological studies, reviews, and systematic reviews. We also excluded studies of developmental disorders and disabilities. The systematic review was pre-registered with PROSPERO: CRD42021292919.

Two investigators (LB and LMJ) independently screened titles and abstracts for inclusion using Covidence. Disagreements were resolved through discussion. Two investigators (LB and AR) extracted information on study characteristics, participant characteristics, pesticide exposure, child development outcomes, and analysis strategy. Data extraction was reviewed by a third investigator (LMJ), and disagreements were resolved through discussion.

We summarised study characteristics narratively. We pooled our results with studies which reported at least one of the same pesticide biomarkers we assessed and at least one child

development domain we assessed. We pooled studies that provided MD or standardised mean difference (SMD) estimates or effect estimates that could be converted to SMD. We made two attempts to contact authors of original studies eligible for the meta-analysis when published information was unavailable or insufficient for pooling. When studies reported estimates for each trimester of pregnancy, we selected the estimate for the first trimester since in our study urine samples were collected primarily during the first trimester (mean gestational age 11.6 ±2.9 weeks). We used a random-effects meta-analysis [41] to estimate summary MDs for the adjusted association between creatinine-adjusted urinary pesticide biomarker concentration and child development composite scores. We assessed heterogeneity between studies using the $I^2$ statistics and statistical significance using the Q statistic [41]. For studies which could not be included in the meta-analysis, we summarised findings narratively.

## 3. Results

### 3.1 Bangladesh cohort

At enrolment, mothers were 23 years of age, on average, 53% had completed secondary school or higher, and 30% of husbands worked in agriculture (**Table 1**). At the 20-40-month follow-up, children were, on average, 26.5 months old (SD 1.9 months). The 284 mothers in our sample were similar to the rest of the enrolment sample (n = 1,329), except that they had higher monthly household income and higher energy and vegetable intake (**S2 Table**). Compared to the rest of the children assessed at the 20-40-month follow-up (n = 532), the 284 children in our sample were younger, had lower development scores (likely because they were younger), and lived in wealthier households (**S3 Table**).

TCPY, a metabolite of chlorpyrifos and chlorpyrifos methyl (organophosphates), was detected in nearly all mothers (98%) and 4-nitrophenol, a metabolite of parathion and methyl parathion (organophosphates), was detected in all mothers (**Table 2**). IMPY, a metabolite of

**Table 1. Characteristics of 284 mother-child pairs in the analytic sample, enrolled in a birth cohort in rural Bangladesh.**

|  | Mean ± SD or N (%) |
| --- | --- |
| *Maternal and household characteristics* |  |
| Age at enrolment, years | 23.1±4.2 |
| Completed secondary school or higher | 149 (52.5) |
| Monthly household income >4000 tk (~$43) | 186 (65.5) |
| Husband engaged in agricultural work | 86 (30.3) |
| *Maternal dietary intake at 28 weeks of gestation* |  |
| Total energy intake (kcal/day) | 3,173.4±734.5 |
| Fruit intake (g/day) | 129.4±64.2 |
| Vegetable intake (g/day) | 161.3±124.7 |
| *Child characteristics* |  |
| Female | 135 (47.5) |
| Age at assessment, months | 26.5±1.9 |
| Cognitive raw score (possible range 0–91) | 59.7±4.1 |
| Receptive communication raw score (possible range 0–49) | 24.1±2.8 |
| Expressive communication raw score (possible range 0–48) | 27.5±4.1 |
| Language raw score (possible range 0–97) | 51.6±6.4 |
| Fine motor raw score (possible range 0–66) | 37.8±1.6 |
| Gross motor raw score (possible range 0–72) | 54.5±2.0 |
| Motor raw score (possible range 0–138) | 92.3±3.0 |

**Table 2. Pesticide biomarker concentrations among 284 pregnant women enrolled in a birth cohort in rural Bangladesh.**

| Pesticide biomarker | >LOD, % (N) [1] | Geometric mean (95% CI), µg/g creatinine[1] | U.S. population, non-pregnant females, geometric mean (95% CI), µg/g creatinine[3] |
|---|---|---|---|
| 2,4-D | 5.6 (16) | - | 0.342 (0.315, 0.372)[4] |
| TCPY | 97.9 (278) | 3.15 (2.79, 3.54) | 0.855 (0.765, 0.954)[5] |
| 4-nitrophenol | 100 (284) | 18.67 (17.02, 20.48) | 0.775 (0.724, 0.827)[5] |
| MDA | 2.8 (8) | - | Not calculated[6] |
| IMPY | 15.8 (45) | - | Not calculated[6] |
| 4-F-3-PBA | 0 (0) | - | Not calculated[6] |
| 3-PBA | 19.4 (55) | - | 0.835 (0.739, 0.942)[4] |
| trans-DCCA | 6 (17) | - | Not calculated[6] |

Abbreviations: 2,4-D, 2,4-dichlorophenoxyacetic acid; TCPY, 3,5,6-trichloro-2-pyridinol; MDA, malathion dicarboxylic acid; IMPY, 2-isopropyl-4-methyl-6-hydroxypyrimidine; 4-F-3-PBA, 4-fluoro-3-phenoxybenzoic acid; 3-PBA, 3-phenoxybenzoic acid; trans-DCCA, trans-3-(2,2-dichlorovinyl)-2,2-dimethylcyclopropane carboxylic acid; LOD, limit of detection

[1] LOD was 0.3 µg/L for 2,4-D, 0.1 µg/L for TCPY, 1.0 µg/L for MDA, 0.6 µg/L for trans-DCCA, and 0.2 µg/L for all other biomarkers.

[2] Geometric mean not reported for pesticide biomarkers detected in <40% of women.

[3] Values are from the National Health and Nutrition Examination Survey, 1999–2018. Centers for Disease Control and Prevention. 2022. Fourth national report on human exposure to environmental chemicals. Updated tables, September 2022. Atlanta, GA: Centers for Disease Control and Prevention. Available at: www.cdc.gov/exposurereport.

[4] Survey years 2013–2014.

[5] Survey years 2009–2010.

[6] Not calculated because proportion of results below LOD was too high to provide a valid result.

diazinon (organophosphate), and 3-PBA, a non-specific metabolite of several pyrethroids, were detected in 16% and 19% of mothers, respectively. Urinary pesticide biomarkers reflect all exposure routes. Because 0% of women in our sample and 30% of their husbands were employed in agriculture and only 1.5% of households in Bangladesh report using indoor residual spraying [42], we hypothesised that dietary intake was the primary pesticide exposure route.

We found that pregnancy concentrations of IMPY were inversely associated with motor scores in the adjusted model, whereas TCPY concentrations were inversely associated with cognitive scores, but the magnitude of the association was small (**Table 3**). Pregnancy concentrations of 4-nitrophenol and 3-PBA were not associated with child development scores. In

**Table 3. Associations between creatinine-adjusted pregnancy pesticide biomarker concentrations (µg/g creatinine) and child development at 20-to-40-months of age, birth cohort in rural Bangladesh[1].**

| | Cognitive composite score | | Language composite score | | Motor composite score | |
|---|---|---|---|---|---|---|
| | Unadjusted MD (95% CI) | Adjusted MD (95% CI) | Unadjusted MD (95% CI) | Adjusted MD (95% CI) | Unadjusted MD (95% CI) | Adjusted MD (95% CI) |
| TCPY | **-0.02 (-0.04, -0.01)** | **-0.02 (-0.04, -0.01)** | 0.00 (-0.02, 0.02) | -0.01 (-0.02, 0.01) | 0.00 (-0.02, 0.01) | -0.01 (-0.02, 0.00) |
| 4-nitrophenol | 0.00 (-0.03, 0.03) | 0.00 (-0.03, 0.02) | -0.02 (-0.06, 0.01) | -0.02 (-0.05, 0.01) | -0.03 (-0.06, 0.00) | -0.02 (-0.04, 0.01) |
| IMPY | -0.31 (-0.98, 0.35) | 0.11 (-0.54, 0.77) | **-0.96 (-1.74, -0.18)** | -0.72 (-1.49, 0.05) | -0.45 (-1.14, 0.23) | **-0.66 (-1.23, -0.09)** |
| 3-PBA | -0.03 (-0.59, 0.53) | 0.16 (-0.39, 0.70) | -0.26 (-0.92, 0.41) | -0.08 (-0.72, 0.57) | 0.10 (-0.48, 0.68) | -0.17 (-0.65, 0.31) |

[1] Estimates significant at 5% level in bold.

Adjusted models control for child age, child sex, maternal age, maternal education, maternal dietary intake, household income, and husband's occupation.

Abbreviations: TCPY, 3,5,6-trichloro-2-pyridinol; IMPY, 2-isopropyl-4-methyl-6-hydroxypyrimidine; 3-PBA, 3-phenoxybenzoic acid; MD, mean difference; CI, confidence interval

sensitivity analyses where TCPY and 4-nitrophenol were specified in terciles and IMPY and 3-PBA as binary variables, none of these pesticide biomarkers were associated with child development, but the direction of the association with IMPY and TCPY was consistently negative (S4 Table).

In exploratory analyses to assess whether adjusted associations differed across child, maternal, and household characteristics, we found that child sex modified the associations between TCPY and language development, IMPY and motor development, and 3-PBA and motor development (S5 Table). Maternal education modified the associations between 4-nitrophenol and motor development. Household income modified the associations between 3-PBA and language and motor development. Although interactions were significant ($p<0.10$), the number of observations in each sub-group was small, leading to limited power and wide CIs. Therefore, we could not determine whether associations were beneficial or harmful among specific sub-groups.

## 3.2 Systematic review and meta-analysis

Of the 1,901 unique records identified, 13 studies were included in this review (Fig 1 and Table 4). Studies were conducted in four countries: China (n = 9) [43–51], Mexico (n = 2) [52, 53], the Philippines (n = 1) [54], and Thailand (n = 1) [55]. Most studies conducted in China came from the Sheyang Mini Birth Cohort Study (n = 5) [43–45, 49, 50] or the Laizhou Wan Birth Cohort (n = 2) [49, 50]. Studies were published between 2011 and 2022. Analytic sample sizes ranged from 82 to 718 (n = 5,111 total participants).

Eleven studies assessed biomarkers in urine [43–51, 53, 55], one in blood [54], and one assessed self-reported exposure [52]. Two studies assessed multiple pesticide types [44, 54]. Six studies assessed organophosphates [43–45, 47, 48, 55], five assessed pyrethroids [44, 46, 51, 53, 54], and three assessed carbamates [49, 50, 54]. One study relying on self-reported exposure did not report specific pesticides [52]. Eight studies assessed motor and language development [43, 45–49, 51, 53, 54], seven assessed adaptive development [43, 45, 47–49, 51, 54], seven assessed personal-social development [43, 45–49, 51], four assessed cognitive development [46, 51–53], two assessed intelligence [44, 50], one assessed performance [54], and one assessed behaviour [55].

Three studies reported on at least one of the same pesticide biomarkers we assessed and on at least one child development domain we assessed, and thus were eligible for pooling [43, 51, 53]. One of these studies, which used a different child development assessment tool than we did (the Gesell Developmental Schedules), provided insufficient information to convert author-reported estimates to MDs or SMDs [43]. A second of these studies classified 3-PBA exposure as <LOD, medium, or high, and provided insufficient information to select a comparable exposure group [53]. No responses were received from the authors of these two studies to requests for data to enable pooling. The third study published sufficient information for pooling associations of 3-PBA, the only common pesticide biomarker between that study and ours [51]. In that study, conducted in Southwest China, 3-PBA was assessed in urine samples from 357 women taken in each trimester of pregnancy (8–12 weeks' gestation, 20–23 weeks' gestation, and 32–35 weeks' gestation). Child development was assessed using BSID-III at 1 year of age [51]. Exposure to 3-PBA during the first or third trimester was not associated with child cognitive, language, motor, socio-emotional, or adaptive development. However, higher exposure during the second trimester was associated with lower cognitive and language scores, but not with motor, socio-emotional or adaptive scores [51]. We selected the first trimester (8–12 weeks' gestation) for pooling since urine samples in our study were collected primarily during the same window (mean gestational age 11.6±2.9 weeks in our study). In the China

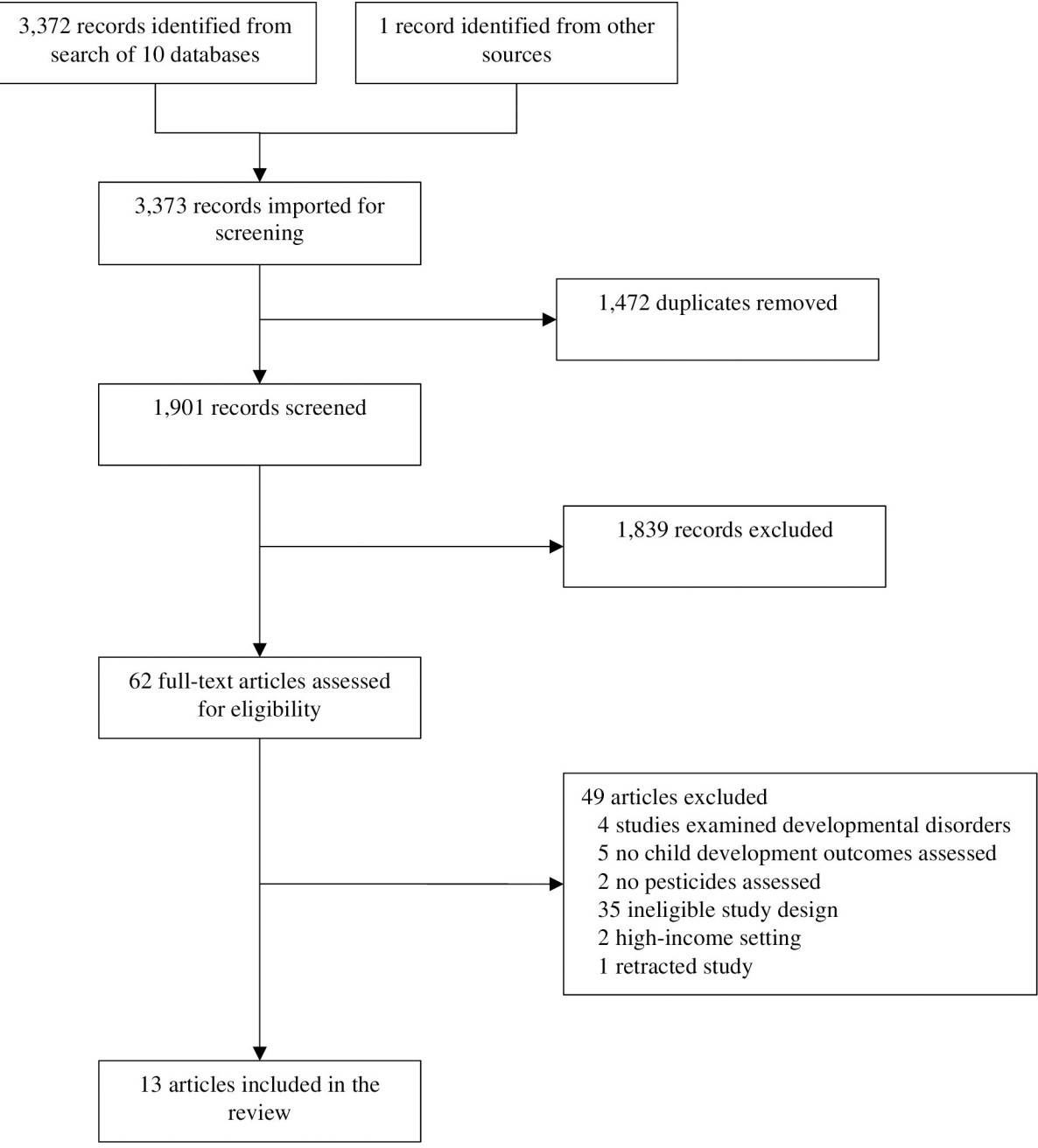

**Fig 1. Preferred Reporting Items for Systematic Reviews and Meta-analyses (PRISMA) flow diagram of search results and included articles.**

study, 3-PBA was detected in 85% of women (geometric mean 2.34 μg/g creatinine) [51]. We summarised estimates for the adjusted association between creatinine-adjusted pregnancy 3-PBA concentrations (μg/g creatinine) and child development composite scores. The pooled results found that pregnancy 3-PBA concentrations were not significantly associated with cognitive (MD 0.11 (95% CI -0.42, 0.64), p = 0.69, $I^2$ = 0.0%, p = 0.43 (**Fig 2**)), language (MD -0.16 (-0.77, 0.45), p = 0.61, $I^2$ = 0.0%, p = 0.47 (**Fig 3**)) or motor composite scores (MD -0.57 (-1.86, 0.72), p = 0.39, $I^2$ = 0.0%, p = 0.16 (**Fig 4**)). The two studies that could not be pooled due to

**Table 4. Characteristics of the prospective studies included in the systematic review of pregnancy pesticide exposure and child development.**

| Author, year | Country | Population assessed | Sample size | Type of exposure | Method of exposure assessment | Time point of exposure assessment | Type of pesticides reported | Domains of child development assessed | Child development assessment tool | Summary of findings | Confounding factors |
|---|---|---|---|---|---|---|---|---|---|---|---|
| Gonzalez-Casanova et al. 2018 [52] | Mexico | Pregnant women enrolled in an Omega-3 Supplementation trial at 18–22 weeks of gestation and their children assessed at 12, 18, 60 and 84 months of age | 718 | Domestic use | Self-reported | 18–22 weeks of gestation | Not reported | Cognitive | Bayley Scales of Infant and Toddler Development, Second Edition SID II (at 12 and 18 months of age); McCarthy Scales of Infant Abilities (at 5 years of age); Wechsler Abbreviated Scale of Intelligence (at 7 years of age) | Pesticide use at home during pregnancy was not associated with average cognitive developmental trajectory: OR 1.28 (95% CI 0.76, 2.15) for average developmental trajectory vs positive developmental trajectory and OR 1.40 (0.79, 2.50) for low developmental trajectory vs positive developmental trajectory. | Socioeconomic status, maternal intelligence, schooling, and supplementation, child sex, breastfeeding status at 3 months of age, home stimulation at 12 months of age, and attendance at private education at 7 years of age |
| Guo et al. 2019 [43] | China | Pregnant women and their children at 3 years of age | 498 | Residential (agricultural region); occupational (agricultural work) | Urine sample | On delivery day | TCPY | Motor, language, personal-social, and adaptive behaviour | Gesell Developmental Schedules | No significant associations between pregnancy TCPY concentrations and child development at 3 years of age: motor: $\beta$ 0.02 (95% CI -0.51, 0.55) language: $\beta$ 0.19 (-0.92, 1.30) personal-social: $\beta$ -0.05 (-0.59, 0.48) adaptive: $\beta$ -0.68 (-1.97, 0.60) No differences between boys and girls. | Maternal education, occupation during pregnancy, family income during pregnancy, urban vs rural residence during pregnancy, parity, passive smoking, child sex and age, season of urine sample collection, cord blood lead levels |
| Guo et al. 2020 [44] | China | Pregnant women and their children at 7 years of age | 347 | Residential (agricultural region); occupational (agricultural work) | Urine sample | On delivery day | 3-PBA, cis-DCCA, trans-DCCA, TCPY | Intelligence | Chinese Revised-Wechsler Intelligence Scale for Children, Fourth Edition | No associations between pregnancy pesticide exposure and child development at 7 years of age:3-PBA: verbal IQ: $\beta$ -0.13 (95% CI -1.47, 1.21) performance IQ: $\beta$ 0.37 (-1.93, 2.67) full IQ: $\beta$ 0.74 (-1.18, 2.67) Σ-DCCA: verbal IQ: $\beta$ -0.09 (-1.28, 1.11) performance IQ: $\beta$ -0.07 (-2.13, 2.00) full IQ: $\beta$ 0.07 (-1.66, 1.79) TCPY: verbal IQ: $\beta$ 0.09 (-1.76, 1.94) performance IQ: $\beta$ 0.31 (-3.50, 2.87) full IQ: $\beta$ -0.58 (-3.24, 2.08) No differences between boys and girls. | Maternal age, education, marital status, and smoking status, family income, breastfeeding duration, child sex, physician for intellectual assessment |
| Liu et al. 2016 [45] | China | Pregnant women and their children at 2 years of age | 310 | Residential (agricultural region); occupational (agricultural work) | Urine sample | On delivery day | DMP, DMTP, DMDTP, DEP, DETP, DEDTP | Motor, language, personal-social, and adaptive behaviour | Gesell Developmental Schedules | Pregnancy DE concentrations were associated with increased risk of being developmentally delayed in the adaptive area: OR 9.75 (95% CI: 1.28, 73.98). This adverse association was observed in boys (OR 26.41 (1.25, 557.40)), but not girls (OR 3.98 (0.20, 77.95)). Pregnancy DE concentrations were not associated with child development in the motor, language, or personal-social areas. Pregnancy DM and DAP concentrations were not associated with child development in any area. | Maternal age, education, occupation, pre-pregnancy BMI, pregnancy weight gain, parity, delivery mode, passive smoking, gestational age, child sex, paternal occupation, family income, cord blood lead value, sampling season, inhabitation during pregnancy |

*(Continued)*

**Table 4.** (Continued)

| Author, year | Country | Population assessed | Sample size | Type of exposure | Method of exposure assessment | Time point of exposure assessment | Type of pesticides reported | Domains of child development assessed | Child development assessment tool | Summary of findings | Confounding factors |
|---|---|---|---|---|---|---|---|---|---|---|---|
| Ostrea et al. 2012 [54] | Philippines | Pregnant women and their children at 2 years of age | 697 | Residential (agro-industrial province) | Maternal hair, maternal blood, infant hair, cord blood, meconium | Mid-gestation (maternal hair and blood) and at birth (cord blood, meconium, infant hair) | Propoxur; pyrethroids (bioallethrin, cyfluthrin, transfluthrin, cypermethrin) | Motor, social and language, performance | Griffiths Mental Developmental Scales | Exposure to propoxur was associated with lower motor development at 2 years of age ($\beta$ -0.14, p<0.001) but was not associated with social or performance development. | Child sex, socio-economic status, maternal intelligence, home stimulation, child blood lead levels |
| Qi et al. 2011 [46] | China | Pregnant women and their children at 1 year of age | 301 | Residential (agricultural province) | Urine sample | During pregnancy | cis-DCCA, trans-DCCA, 3-PBA | Motor, social, and mental | Developmental Screening Test | Pregnancy pyrethroid exposure was negatively associated with neurodevelopment ($\beta$ -0.145, p<0.05) | Maternal education, child's place of residence, primary caregiver, and post-birth illness |
| Qi et al. 2022 [51] | China | Pregnant women and their children at 1 year of age | 419 | Not specified | Urine sample | First (8–12 weeks of gestation), second (20–23 weeks of gestation) and third (32–35 weeks of gestation) trimester | 3-PBA, 4 F-3-PBA, cis-DBCA | Cognitive, motor, language, socio-emotional, adaptive | Bayley Scales of Infant and Toddler Development, Third Edition | Pregnancy exposure to 3-PBA in the second trimester was associated with lower cognitive ($\beta$ -3.34 (95% CI -6.11, -0.57) and language ($\beta$ -2.90 (-5.20, -0.61) development. Pregnancy exposure to cis-DBCA in the second trimester was associated with lower adaptive behaviour ($\beta$ -0.73 (-1.27, -0.19). Pregnancy exposure to 4 F-3-PBA and cis-DBCA in the third trimester was associated with higher language and adaptive behaviour scores ($\beta$ 6.04 (1.84, 10.23) and $\beta$ 0.73 (0.29, 1.17), respectively). | Maternal age, education, poverty, perceived stress, weight gain, urine cotinine concentration during pregnancy, child sex, birthweight z-scores, parenting time for children, primary caregiver, breastfeeding, passive smoking |
| Wang et al. 2017 [47] | China | Pregnant women and their children assessed at 12 and 24 months of age | 436 | Residential (proximity to pesticide factories), household insecticide use, food residues | Urine sample | On delivery day | DEDTP, DETP, DEP, DMDTP, DMTP, DMP | Motor, language, personal-social, and adaptive behaviour | Gesell Developmental Schedules | A 10-fold increase in pregnancy Des and DAPs was associated with a -2.59-point (95% CI -4.71, -0.46) and -2.49-point (-4.85, -0.14) decrease in social development at 24 months of age. This inverse association between Des and social development was observed in boys (-3.20 (-6.31, -0.10), but not in girls (-1.59 (-4.53, 1.35)). No significant associations between pregnancy DMs, Des, or DAPs and child development at 12 months of age or between DMs and child development at 24 months of age. | Child sex, household income, paternal education, smoking during pregnancy, maternal education, IQ, and age |
| Wang et al. 2020 [48] | China | Pregnant women and their children assessed at 12 and 24 months of age | 436 | Residential (proximity to pesticide factories), household insecticide use, food residues | Urine sample | On delivery day | DEDTP, DETP, DEP, DMDTP, DMTP, DMP | Motor, language, personal-social, and adaptive behaviour | Gesell Developmental Schedules | A 10-fold increase in pregnancy DMs was associated with a -5.72-point (95% CI -11.29, -0.16) decrease in social development at 24 months of age among children of mothers carrying PON1$_{-108}$CC. A 10-fold increase in pregnancy DMs and DAPs was associated with a -7.68-point (-13.91, -1.46) and a -7.67-point (-15.06, -0.27) decrease, respectively, in gross motor development at 24 months of age among children of mothers carrying PON1$_{192}$QQ. | Birth weight, maternal age, smoking during pregnancy, child sex, household income, parental education |

(Continued)

**Table 4.** (Continued)

| Author, year | Country | Population assessed | Sample size | Type of exposure | Method of exposure assessment | Time point of exposure assessment | Type of pesticides reported | Domains of child development assessed | Child development assessment tool | Summary of findings | Confounding factors |
|---|---|---|---|---|---|---|---|---|---|---|---|
| Watkins et al. 2016 [53] | Mexico | Pregnant women and their children assessed at 24 and 36 months of age | 187 | Dietary, residential, and domestic exposure hypothesized, but not empirically confirmed | Urine sample | Third trimester | 3-PBA | Mental, psychomotor | Bayley Scales of Infant and Toddler Development, Second Edition | Higher pregnancy exposure to 3-PBA was associated with lower mental development at 24 months of age: -3.5 and -3.8 points for medium and high categories, respectively, relative to low/non-detectable category. These associations were significant in girls, but not in boys. Pregnancy 3-PBA levels were not associated with mental development at 36 months of age or with psychomotor development at 24 or 36 months of age. | Maternal IQ, education, socio-economic status, blood lead level, urinary specific gravity, child sex |
| Woskie et al. 2017 [55] | Thailand | Pregnant women and their children assessed at birth | 82 | Occupational (agricultural worker or living with an agricultural worker) | Urine sample | At 6 months of gestation and at birth | DMP, DEP, DETP, DEDTP | Behaviour | Brazelton Neonatal Behavioural Assessment Scale | Higher DMP levels were associated with higher NBAS Habituation cluster score: β 1.74 (95% CI 0.11, 3.35). Higher DEP and total DEP levels were associated with higher NBAS Range of State score: β 0.16 (0.003, 0.31) and β 0.23 (0.05, 0.41), respectively. Being an agricultural worker during pregnancy was not associated with NBAS scores. | Habituation : NBAS tester, parity Orientation : NBAS tester, parity Motor performance: NBAS tester, self-reported income sufficiency Range of state: NBAS tester, maternal education, marital status, alcohol use, cough medicine use Regulation of state: NBAS tester, marital status, alcohol use, maternal age Autonomic stability: NBAS tester, marital status, cough medicine use Number of abnormal reflexes: NBAS tester, caffeine use |
| Zhang et al. 2019 [49] | China | Pregnant women and their children assessed at 3 years of age | 377 | Residential (agricultural region); occupational (agricultural work) | Urine sample | On delivery day | Carbofuranphenol | Motor, language, personal-social, and adaptive behaviour | Gesell Developmental Schedules | Higher pregnancy carbofuranphenol levels were associated with lower adaptive (β -0.755 (95% CI -1.257, -0.254)), social (β -0.341 (-0.656, -0.027)), and total development (β -0.349 (-0.693, -0.005)) at 3 years of age. Lower adaptive development was observed in girls (β -0.693 (-1.326, -0.059)), but not in boys (β 0.136 (-0.213, 0.486)). | Maternal age, education, household income, family urban vs rural location, passive smoking |
| Zhang et al. 2020 [50] | China | Pregnant women and their children assessed at 7 years of age | 303 | Residential (agricultural region); occupational (agricultural work) | Urine sample | On delivery day | Carbofuranphenol | Intelligence | Chinese Revised-Wechsler Intelligence Scale for Children, Fourth Edition | Pregnancy carbofuranphenol levels were not associated with verbal, performance or full-scale IQ at 7 years of age. | Maternal age, education, paternal education, singleton pregnancy, child sex, age at assessment, family income, child development assessor |

Abbreviations: BMI, body mass index; CI, confidence interval; DBCA, 3-(2,2-dibromovinyl)-2,2-dimethylcyclopropane carboxylic acid; DCCA, 3-(2,2-dichlorovinyl)-2,2-dimethylcyclopropane carboxylic acid; DAP, dialkylphosphate; DE, diethylphosphate; DEDTP, diethyldithiophosphate; DETP, diethylthiophosphate; DM, dimethylphosphate; DMDTP, dimethydithiophosphate; DMP, dimethylphosphate; DMTP, dimethylthiophosphate; NBAS, Neonatal Behavioural Assessment Scale; OR, odds ratio; 3-PBA, 3-phenoxybenzoic acid; PON1, Paraoxonase 1; TCPY, 3,5,6-trichloro-2-pyridinol

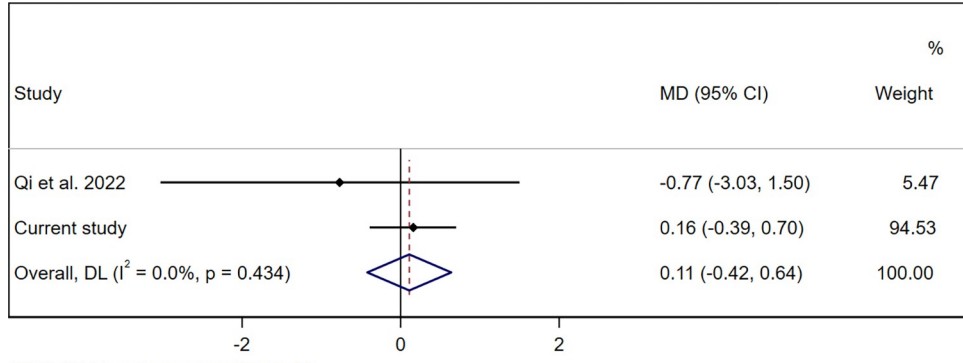

**Fig 2. Pooled association between creatinine-adjusted pregnancy 3-PBA concentrations (μg/g creatinine) and child cognitive development composite scores at 20-to-40 months of age.** Abbreviations: 3-PBA, 3-phenoxybenzoic acid; MD, mean difference; CI, confidence interval.

lack of comparable information or raw data found that pregnancy concentrations of 3-PBA were associated with poorer mental development at 24 months of age [53], but not at 36 months or with motor development at 24 or 36 months [53] or IQ at 7 years [44].

Other included studies found inconsistent associations between pregnancy concentrations of *cis*-DCCA and *trans*-DCCA and cognitive, language, and adaptive development at 1 year of age and IQ at 7 years of age [44, 46, 51]. Associations between pregnancy concentrations of diethylphosphate, dimethylphosphate, and dialkylphosphate and child motor and language development at 24 months of age were also inconsistent [45, 47, 48]. Pregnancy dimethylphosphate and diethylphosphate concentrations were inversely associated with infant behaviour [55], whereas pregnancy concentration of TCPY was not associated with child development at 3 or 7 years of age [43, 44]. Lastly, pregnancy concentrations of propoxur and carbofuranphenol (metabolites of carbamate pesticides) were associated with lower motor development at 2 years of age [54] and lower adaptive and social development at 3 years of age [49], but not with IQ at 7 years of age [50].

## 4. Discussion

In this prospective analysis using data from a birth cohort in rural Bangladesh, we found that pregnancy concentrations of IMPY, an organophosphate biomarker, were inversely associated

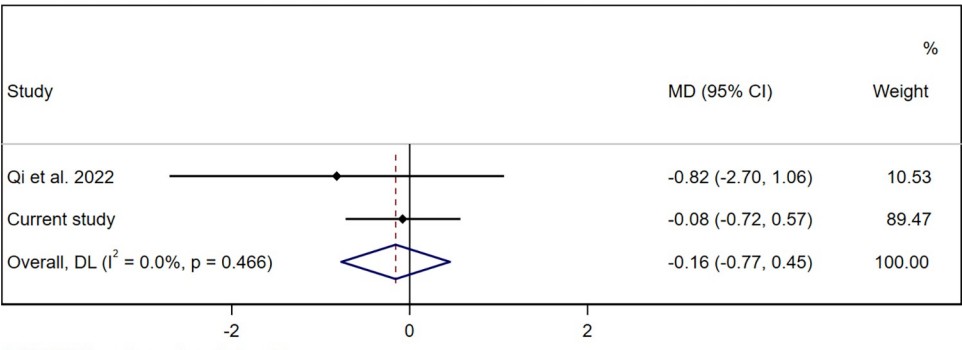

**Fig 3. Pooled association between creatinine-adjusted pregnancy 3-PBA concentrations (μg/g creatinine) and child language development composite scores at 20-to-40 months of age.** Abbreviations: 3-PBA, 3-phenoxybenzoic acid; MD, mean difference; CI, confidence interval.

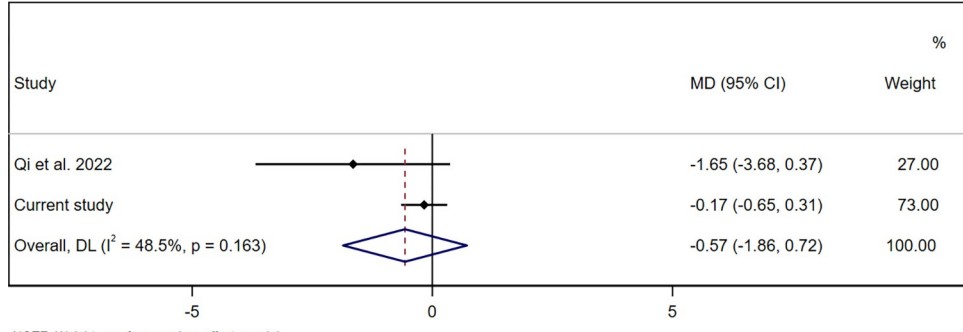

**Fig 4. Pooled association between creatinine-adjusted pregnancy 3-PBA concentrations (μg/g creatinine) and child motor development composite scores at 20-to-40 months of age.** Abbreviations: 3-PBA, 3-phenoxybenzoic acid; MD, mean difference; CI, confidence interval.

with motor scores among 20-40-month-old children in rural Bangladesh, but pregnancy concentrations of 3-PBA (a metabolite of several pyrethroid insecticides) were not. With respect to cognitive development, only pregnancy TCPY concentrations (an organophosphate insecticide metabolite) were inversely associated with cognitive scores, but the association was very small and not clinically meaningful. Overall, these small associations were not unexpected given the complex aetiology of child development, where many factors can play a role [56], and exposure misclassification, which would bias estimates towards the null and result in smaller associations. Limited detection of biomarkers and inadequate exposure assessment at a single point during pregnancy could also help explain these small associations. These findings were supported by our systematic review which found inconsistent associations between pregnancy exposure to organophosphates, pyrethroids, and carbamates and child development up to 7 years of age. Our meta-analysis which pooled our data with one other study found that 3-PBA was not associated with cognitive, language, or motor development. Results from the Bangladesh cohort also found that child, maternal, and household characteristics modified the associations between pregnancy pesticide biomarker concentrations and child language and motor development. Prior evidence is inconsistent on whether child sex modifies the associations between pregnancy pesticide biomarkers and child development [43–45, 47, 49, 53]. Further unpacking associations by subgroups can help inform the targeting of interventions to reduce pesticide exposure and improve child development.

Several mechanisms can explain the associations between pregnancy pesticide biomarkers and child development, including inhibition of AChE activity, brain anomalies such as cortical thinning and regional enlargement of white matter, and changes in the function of the nervous system such as altered electrophysiological function of the sensory, visual, and auditory cortex or disrupted transduction signalling function [16, 17, 21–23, 57]. Evidence to-date suggests these mechanisms are not sex-specific [16]. Given the plausibility of these mechanisms, limitations of our analysis and existing literature can likely explain the inconsistent associations we observed. First, two important toxicology limitations should be noted. One is that all studies, including ours, assessed a few active ingredients. For example, the included studies conducted in China examined up to six active pesticide ingredients, while 239 active ingredients are currently approved in China [30]. Further, people are usually exposed to multiple active ingredients or complex mixtures, which were not examined in ours or other included studies. The second toxicology limitation is that most of the biomarkers assessed in the included studies were developed for use in HICs and do not match to frequently used active ingredients in LMICs. As evidenced in our sample, three of the pesticide biomarkers we quantified were not

detected in most participants and one was not detected in any participants. From the four pesticide biomarkers we quantified, IMPY and 3-PBA were detected in fewer than 20% of women. Therefore, our results should be interpreted with caution.

Another important limitation is that most studies, including ours, assessed pesticide exposure at one time point during pregnancy with most studies collecting one spot urine sample on delivery day. Only one study assessed pesticide exposure in each trimester and found that associations varied by trimester, with the first and second trimester being particularly sensitive windows [51]. Relatedly, most studies, including ours, assessed child development at a single time point in early childhood (1–3 years of age). Given the brain's plasticity in early life and potential for neuronal compensation [24], this length of follow-up might be insufficient for the effects of long-latency delayed neurotoxicity to manifest, which can explain the null findings we and others have observed. Relatedly, given the wide use of pesticides in Bangladesh [3], it is likely that children were exposed to pesticides postnatally. Unfortunately, data on postnatal exposure were not available in this cohort, and so there is the potential for residual confounding of the associations between prenatal exposure and child development reported here.

Lastly, nine of the 13 studies we included in the systematic review were conducted in China, and most of these studies drew their samples from two birth cohorts. We did not identify any articles from Sub-Saharan Africa, South America, or Asia Pacific. Because of insufficient information in the published articles and because authors did not provide responses to requests for additional information, we were only able to pool our estimates with one other study. Therefore, our findings may have limited generalisability to other LMICs.

Despite these limitations, existing literature suggests that pregnancy exposure to some organophosphate pesticides is associated with poorer child development in certain domains. In our Bangladesh cohort and three other studies included in the systematic review [47, 48, 53], dietary intake was the primary hypothesised route of exposure. In the remaining studies, the primary route of exposure was residential (living in an agricultural region) or occupational (agricultural worker) [43–50, 54, 55]. Given these exposure routes, several potential interventions to reduce pesticide exposure and improve child development are worth noting. First, consumption of organic foods can reduce pregnancy pesticide exposure through dietary intake. One study of 20 pregnant women in the United States found that substituting conventional for organic fruit and vegetables reduced pregnancy exposure to some pesticides [58]. To the best of our knowledge, no similar interventions promoting organic foods have been evaluated among pregnant women in LMICs. The feasibility and cost of such organic feeding interventions in LMICs would benefit from further evaluation given issues around availability, accessibility, and affordability of organic foods across different socio-economic groups in LMICs.

Adopting organic farming is another intervention which can reduce pesticide exposure through dietary intake (by increasing organic foods availability), residential proximity (by reducing pesticide use on farms), and occupation (by reducing pesticide use by farmers). Farmers in LMICs, including Bangladesh, generally have positive attitudes towards organic farming [59–61], indicating that organic farming interventions could be an acceptable strategy to reduce pesticide exposure. However, such interventions should consider the local context and address context-specific bottlenecks to organic farming. For example, in Bangladesh, only 0.1% of agricultural land is dedicated to organic farming largely due to issues with land ownership, credit access, and market access [62]. Given these challenges with organic farming, other approaches to reducing pesticide use/misuse in agriculture such as integrated pest management, government training and inspection of pesticide retailers, agricultural extension support to farmers, and bans on the production and import of highly hazardous pesticides should be explored.

In contexts where interventions to directly reduce pesticide exposure are infeasible or slow to initiate and generate change, protective approaches such as responsive stimulation and parenting interventions or maternal folate intake may help offset the negative effects of pesticide exposure on child development. Responsive stimulation and parenting interventions are effective in improving the caregiving environment and in turn child development [63], including those tested in Bangladesh [64–66]. However, evidence on if, and how, these types of interventions mediate the effects of pesticide exposure on child development is lacking. Further work can help to understand and test the potential synergistic or additive effects on child development of combining interventions to reduce pesticide exposure and stimulation interventions. With respect to maternal folate intake, evidence from high-income countries suggests that maternal folate intake during pregnancy can be protective against developmental neurotoxicity, particularly in children with developmental disorders [67, 68]. For example, one study in the United States found that high folic acid intake ($\geq$800 μg) early in pregnancy was associated with lower odds of autism spectrum disorder at age 2–5 years [67]. While more research is needed to confirm these findings, continuing to recommend that pregnant women consume iron-folic acid supplements [69] may have co-benefits by attenuating the adverse consequences of pesticide exposure on child development.

## 5. Conclusion

Pregnancy urinary pesticide concentrations of two biomarkers of organophosphates were inversely associated with cognitive and motor development scores among 20-40-month-old children in rural Bangladesh, while pregnancy concentrations of 4-nitrophenol and of a non-specific metabolite of pyrethroids were not. Our systematic review included 13 studies from four LMICs and found inconsistent associations between pregnancy exposure to organophosphates, pyrethroids, and carbamates and child development up to 7 years of age. In analyses which pooled our study with another study, pregnancy concentrations of 3-PBA (a pyrethroid insecticide metabolite) were not associated with cognitive, language, or motor development in early childhood. Our findings suggest that additional research can increase the understanding on whether pregnancy pesticide exposure influences child development across the life course.

## CDC disclaimer

The findings and conclusions of this report are those of the authors and do not necessarily represent the official position of the Centers for Disease Control and Prevention (CDC). Use of trade names is for identification only and does not imply endorsement by the CDC, the Public Health Service, or the U.S. Department of Health and Human Services.

## Supporting information

**S1 Checklist. PRISMA 2020 for abstracts checklist.**
(DOCX)

**S2 Checklist. PRISMA 2020 checklist.**
(DOCX)

**S3 Checklist. STROBE statement—Checklist of items that should be included in reports of *cohort studies*.**
(DOCX)

**S1 Table. Search terms used in PubMed.**
(DOCX)

**S2 Table. Comparison of enrolment characteristics of mother-child pairs with pesticide data included in the analytic sample and mother-child pairs without pesticide data excluded from the analysis, birth cohort in rural Bangladesh.**
(DOCX)

**S3 Table. Comparison of characteristics at follow-up of mother-child pairs with pesticide data included in the analytic sample and mother-child pairs without pesticide data excluded from the analysis, among the sub-sample of children assessed on the Bayley Scales of Infant and Toddler Development, birth cohort in rural Bangladesh.**
(DOCX)

**S4 Table. Associations between creatinine-adjusted prenatal pesticide biomarker concentrations (μg/g creatinine) and child development at 20-to-40-months of age, birth cohort in rural Bangladesh.**
(DOCX)

**S5 Table. Heterogeneity of the adjusted associations between creatinine-adjusted prenatal pesticide biomarker concentrations (μg/g creatinine) and child development at 20-to-40-months of age by child sex, maternal education, household income, and husband's occupation, birth cohort in rural Bangladesh.**
(DOCX)

## Acknowledgments

We would like to thank Dr David C Bellinger for his role in the study design, training of study staff, and quality control of the child development assessment. We are grateful to Dr Yumei Hseuh's laboratory at Taipei Medical University for storing the urine samples and measuring creatinine, and Dr Dickson Wambua, Mr William Roman, Mr Isuru Vidanage, and Ms Meghan Vidal for the quantification of pesticides biomarkers. We would also like to thank Dr Quazi Quamruzzamane for his role in the study design, and supervision of data collection and clinical operations in Bangladesh. We thank Dr Robert Wright for this participation in the study design and Md Omar Sharif Ibne Hasan for performing the neurological testing. We are grateful to Ms Ying Dong for her assistance with data extraction from articles published in Chinese for the systematic review.

## Author Contributions

**Conceptualization:** Lilia Bliznashka, David C. Christiani, Maitreyi Mazumdar, Lindsay M. Jaacks.

**Data curation:** Lilia Bliznashka, Nancy Diao.

**Formal analysis:** Lilia Bliznashka, Aditi Roy, Antonia M. Calafat, Maria Ospina, Lindsay M. Jaacks.

**Funding acquisition:** David C. Christiani, Maitreyi Mazumdar, Lindsay M. Jaacks.

**Investigation:** Lilia Bliznashka, Aditi Roy, David C. Christiani, Maitreyi Mazumdar, Lindsay M. Jaacks.

**Methodology:** Lilia Bliznashka, Aditi Roy, Lindsay M. Jaacks.

**Project administration:** David C. Christiani, Maitreyi Mazumdar.

**Supervision:** Lindsay M. Jaacks.

**Validation:** Aditi Roy, Lindsay M. Jaacks.

**Visualization:** Lilia Bliznashka.

**Writing – original draft:** Lilia Bliznashka.

**Writing – review & editing:** Lilia Bliznashka, Aditi Roy, David C. Christiani, Antonia M. Calafat, Maria Ospina, Nancy Diao, Maitreyi Mazumdar, Lindsay M. Jaacks.

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
