## [Decision Letter · Decision Letter 0]

29 Mar 2023

PONE-D-23-02593Pregnancy pesticide exposure and child development in low- and middle-income countries: a prospective analysis of a birth cohort in rural Bangladesh and meta-analysisPLOS ONE

Dear Dr. Bliznashka,

Thank you for submitting your manuscript to PLOS ONE. After careful consideration, we feel that it has merit but does not fully meet PLOS ONE’s publication criteria as it currently stands. Therefore, we invite you to submit a revised version of the manuscript that addresses the points raised by two reviewers.

 Please submit your revised manuscript by May 13 2023 11:59PM. If you will need more time than this to complete your revisions, please reply to this message or contact the journal office at plosone@plos.org. Please include the following items when submitting your revised manuscript:A rebuttal letter that responds to each point raised by the academic editor and reviewer(s). You should upload this letter as a separate file labeled 'Response to Reviewers'.A marked-up copy of your manuscript that highlights changes made to the original version. You should upload this as a separate file labeled 'Revised Manuscript with Track Changes'.An unmarked version of your revised paper without tracked changes. You should upload this as a separate file labeled 'Manuscript'.

We look forward to receiving your revised manuscript.

Kind regards,

Iman Al-Saleh

Academic Editor

PLOS ONE

“Funding for the prospective study was provided by the Burke Global Health Fellowship program at the Harvard Global Health Institute and the National Institutes of Health (R01-ES015533, P30-ES00002, P42-ES016454). LB and LMJ were supported by the Medical Research Council/UK Research and Innovation (Grant Ref: MR/T044527/1). AR was supported by a DBT/Wellcome Trust India Alliance Fellowship (grant number IA/CPHE/20/1/505272). For the purpose of open access, the author has applied a Creative Commons Attribution (CC BY) licence to any Author Accepted Manuscript version arising from this submission.”

**Reviewers' comments:**

Reviewer's Responses to Questions

**Comments to the Author**

1. Is the manuscript technically sound, and do the data support the conclusions?

Reviewer #1: Yes

Reviewer #2: Partly

2. Has the statistical analysis been performed appropriately and rigorously? 

Reviewer #1: Yes

Reviewer #2: Yes

3. Have the authors made all data underlying the findings in their manuscript fully available?

Reviewer #1: Yes

Reviewer #2: Yes

4. Is the manuscript presented in an intelligible fashion and written in standard English?

Reviewer #1: Yes

Reviewer #2: Yes

5. Review Comments to the Author

**Reviewer #1**: The authors present a thorough and well-written assessment of prenatal pesticide exposures on early child development by conducting a primary analysis on 284 mother-child pairs and a systematic review and meta-analysis. Overall, the methods and conclusions of this paper are sound. I have some comments that I believe could help strengthen this manuscript:

Line 10: Include the average GA at urine sample collection.

Line 25: Given concerns regarding exposure misclassification of 3-PBA due to collection of a single time point along with the rapid half-lives of pyrethroids, I would change “we confirmed that pregnancy…” to something along the lines of “we found consistent evidence of…”

The introduction is beautifully written but based on the discussion of a pyrethroid in the abstract, I expected some mention of pyrethroids in the introduction, even if just a sentence or two.

Line 120: What was the selection criteria for adjustment covariates? Was it based on prior literature? If so, I would cite the literature used to compile this list. If based on a Directed Acyclic Graph, I would include it in a supplement. Additionally, when was information on study covariates measured? Was it at study entry? I would mention that in the text.

Line 294-296: I would directly state here that exposure misclassification is a concern as well which would bias results towards the null, resulting in smaller associations.

Line 307-309: It might be more informative to provide explicit examples of impacts to the functioning of the nervous system and other mentioned biological systems. Additionally, are there sex-specific biological mechanisms, given the sex-specific interactions observed?

Although the purpose of this analysis was to evaluate the effects of prenatal exposures on the development of 20 to 40-month-old children, it is possible that postnatal exposures could be associated with both prenatal exposures and child development and could thus be a possible confounder. If this data is available, it would be a good sensitivity analysis but if not available, it should be mentioned as a potential limitation.

**Reviewer #2**: General comments

Overall, the site-specific portion of the paper was well done and informative. However, the results of the systematic review portion was overstated. The authors proudly state that their systematic review included 13 papers, but their pooled analysis only involved pooling with one additional study. For 3PBA, the summary tables provided indicate that 4 papers had evaluations with 3PBA, and that of those 4 papers, 3 found an adverse association with 3PBA and development, and 1 found no association. The authors then pooled with one data set that found an association, but when combined with the data from the site-specific paper, no longer saw an association. Also of note was that fact that the site specific paper only had 20% of participants with detectable levels of 3BPA. I hardly think this constitutes enough evidence to state “We confirmed that pregnancy 3-PBA concentrations were not associated with cognitive, language, or motor development.

Specific Comments

Abstract: Revise language on “confirming findings” as listed above.

Introduction: Line 39 – the sentence on adverse health effects in adults is vague and should be revised.

Methods: Line 111 – Videotaping the evaluations for quality control was mentioned, but there never seemed to be any information on if the quality was found to be consistent. I may have missed it. Seems like something that could be added to the supplemental material, but if you are going to highlight that you did it, results need to be somewhere.

Results:

Lines 205-209 – It is mentioned that there was some evidence of associations with various factors such as sex and SES, but that nothing could be determined from them. However, the modification analysis is brought up again in the discussion. The authors either need to bring more information on what they found to the main text, or remove the bit on modification from the discussion.

Lines 240-262 – This section is not presented in a clear way. First, all of the studies considered are listed. Then, the list is reduced to just what is available for pooling, which is three potential studies. Then suddenly we are down to one study, one compound. The only findings of the other studies that are presented are those from the two considered for pooling that could not be pooled. At least some minimal comments need to be made on the overall findings related to 3PBA. And also, there should be more about the study they did pool with, what were the findings, how did the levels of exposure compare?

Discussion:

Lines 296-298 – If part of the project was to do a meta analysis, shouldn’t the information on what was found in other OP studies be included in the results?

Lines 298-300 – instead of “showed”, “found no association”, evidence is not strong enough to “show” anything.

Line 300-306 – Since no information on the modifications were presented in the results, this is really hard to interpret.

Lines 351-360 – Before jumping straight to organic, it might make sense to work on programs to at least get usage down since they seem to be clearly overusing the chemical. It is likely much easier to at least come in line with Western usage than jump straight to organic. Also, you have a typo with your ().

Lines 361-369 – I don’t recall the literature on pesticides and folate intake, but I recall there might be some findings showing less of an impact when moms take folate right from the beginning. If you are going to speculate on potential interventions, this is a well proven one from a public health standpoint and is very low cost. Could be good to mention in a sentence or two.

Limitations: I think the following limitation needs to be mentioned somewhere. These woman were exposed to massive levels of organophosphate pesticides. It is not at all surprising that they could not find a relationship with a class of compounds that does not appear to be regularly used in Bangladesh (only 20% detects).

Conclusion: lines 374-377 – you can’t jump from 13 studies to “pooled analyses” when you only pooled with one study. See general comments

6. PLOS authors have the option to publish the peer review history of their article (what does this mean?). If published, this will include your full peer review and any attached files.

Reviewer #1: No

Reviewer #2: No

---

## [Author Response · Author response to Decision Letter 0]

3 May 2023

We thank the reviewers for their time and careful consideration of our manuscript. We have incorporated their suggestions, which we believe have improved the paper. We provide point-by-point responses below highlighting changes made in the text. Line numbers below refer to the clean version of the manuscript (without tracked changes). 

REVIEWER #1

The authors present a thorough and well-written assessment of prenatal pesticide exposures on early child development by conducting a primary analysis on 284 mother-child pairs and a systematic review and meta-analysis. Overall, the methods and conclusions of this paper are sound. I have some comments that I believe could help strengthen this manuscript:

Thank you for your time and support of our manuscript.

Line 10: Include the average GA at urine sample collection.

Thank you for this suggestion. We have specified on lines 10-11: “Eight urinary pesticide biomarkers were quantified in early pregnancy (mean gestational age 11.6±2.9 weeks) as an index of pesticide exposure.”

Line 25: Given concerns regarding exposure misclassification of 3-PBA due to collection of a single time point along with the rapid half-lives of pyrethroids, I would change “we confirmed that pregnancy…” to something along the lines of “we found consistent evidence of…”

Thank you for this suggestion. We have revised accordingly (lines 25-27): “After pooling our results with one other study, we found consistent evidence that pregnancy 3-PBA concentrations were not associated with cognitive, language, or motor development.”

The introduction is beautifully written but based on the discussion of a pyrethroid in the abstract, I expected some mention of pyrethroids in the introduction, even if just a sentence or two.

Thank you for this suggestion. We have now noted in the Introduction that pyrethroid food residues are an issue in Bangladesh (lines 36-39): “In some parts of the country, pesticide residues, including organophosphates and pyrethroids, are frequently detected in vegetables and fruit sold in markets [6–9] High concentrations of organophosphate, pyrethroid, and carbamate residues are also frequently found in water and soil [10–12].” 

We have also summarised a recent systematic review on pyrethroids and child development (lines 63-66): “Yet another recent review concluded that there was sufficient evidence of an adverse association between prenatal pyrethroid exposure and child neurodevelopment, based on 17 studies only four of which were in LMICs: China (n=2), Mexico (n=1), and South Africa (n=1) [29].”

Line 120: What was the selection criteria for adjustment covariates? Was it based on prior literature? If so, I would cite the literature used to compile this list. If based on a Directed Acyclic Graph, I would include it in a supplement. Additionally, when was information on study covariates measured? Was it at study entry? I would mention that in the text.

Thank you for the suggestion. We previously developed a Directed Acyclic Graph (DAG) to evaluate the association between maternal pesticide exposure and child growth in this cohort (see supplement to Jaacks et al. 2019 Environment International). This DAG was used to identify covariates for this analysis of maternal pesticide exposure and child development. We have clarified on lines 123-128: “Adjusted estimates controlled for an a priori set of confounders, selected using a previously published Direct Acyclic Graph [33]: child age at assessment, child sex, maternal age at enrolment, maternal education at enrolment, and maternal energy, vegetable, and fruit intake (assessed using a semi-quantitative food frequency questionnaire [39] administered at 28 weeks’ gestation), husband’s occupation at enrolment (agricultural work vs. not), and household income at enrolment.”

Line 294-296: I would directly state here that exposure misclassification is a concern as well which would bias results towards the null, resulting in smaller associations.

Thanks, we have noted the concern (lines 317-319): “Overall, these small associations were not unexpected given the complex aetiology of child development, where many factors can play a role [56], and exposure misclassification, which would bias estimates towards the null and result in smaller associations.”

Line 307-309: It might be more informative to provide explicit examples of impacts to the functioning of the nervous system and other mentioned biological systems. Additionally, are there sex-specific biological mechanisms, given the sex-specific interactions observed?

Thank you for this suggestion. We have expanded on the biological mechanisms (lines 332-337): “Several mechanisms can explain the associations between pregnancy pesticide biomarkers and child development, including inhibition of AChE activity, brain anomalies such as cortical thinning and regional enlargement of white matter, and changes in the function of the nervous system such as altered electrophysiological function of the sensory, visual, and auditory cortex or disrupted transduction signalling function [16,17,21–23,57]. Evidence to-date suggests these mechanisms are not sex-specific [16].”

Although the purpose of this analysis was to evaluate the effects of prenatal exposures on the development of 20 to 40-month-old children, it is possible that postnatal exposures could be associated with both prenatal exposures and child development and could thus be a possible confounder. If this data is available, it would be a good sensitivity analysis but if not available, it should be mentioned as a potential limitation.

Thank you for noting this limitation, which we have now acknowledged (lines 358-361): “Relatedly, given the wide use of pesticides in Bangladesh [3], it is likely that children were exposed to pesticides postnatally. Unfortunately, data on postnatal exposure were not available in this cohort, and so there is the potential for residual confounding of the associations between prenatal exposure and child development reported here.”

 

REVIEWER #2

Overall, the site-specific portion of the paper was well done and informative. However, the results of the systematic review portion was overstated. The authors proudly state that their systematic review included 13 papers, but their pooled analysis only involved pooling with one additional study. For 3PBA, the summary tables provided indicate that 4 papers had evaluations with 3PBA, and that of those 4 papers, 3 found an adverse association with 3PBA and development, and 1 found no association. The authors then pooled with one data set that found an association, but when combined with the data from the site-specific paper, no longer saw an association. Also of note was that fact that the site specific paper only had 20% of participants with detectable levels of 3BPA. I hardly think this constitutes enough evidence to state “We confirmed that pregnancy 3-PBA concentrations were not associated with cognitive, language, or motor development.

We thank the reviewer for their time and thoughtful comments. We agree that we may have overstated the findings from the systematic review portion and have toned down the language throughout the manuscript. 

Specific Comments

Abstract: Revise language on “confirming findings” as listed above.

Thank you, based on feedback from both reviewers we have removed the word ‘confirming’ and revised to: “After pooling our results with one other study, we found consistent evidence that pregnancy 3-PBA concentrations were not associated with cognitive, language, or motor development.” (lines 25-27)

Introduction: Line 39 – the sentence on adverse health effects in adults is vague and should be revised.

Thank you for this suggestion. We have clarified (lines 40-41): “Widespread exposure to pesticides results in numerous carcinogenic, reproductive, immunological, neurological, and other adverse health effects in adults [13,14].”

Methods: Line 111 – Videotaping the evaluations for quality control was mentioned, but there never seemed to be any information on if the quality was found to be consistent. I may have missed it. Seems like something that could be added to the supplemental material, but if you are going to highlight that you did it, results need to be somewhere.

Thank you for this suggestion. Unfortunately, we no longer have the data from this quality control, and we have therefore removed the statement from the manuscript. 

Results: 

Lines 205-209 – It is mentioned that there was some evidence of associations with various factors such as sex and SES, but that nothing could be determined from them. However, the modification analysis is brought up again in the discussion. The authors either need to bring more information on what they found to the main text, or remove the bit on modification from the discussion.

Thank you for this suggestion. We have brought more information on the results in the main text (lines 211-220): “In exploratory analyses to assess whether adjusted associations differed across child, maternal, and household characteristics, we found that child sex modified the associations between TCPY and language development, IMPY and motor development, and 3-PBA and motor development (Table in S5 Table). Maternal education modified the associations between 4-nitrophenol and motor development. Household income modified the associations between 3-PBA and language and motor development. Although interactions were significant (p<0.10), the number of observations in each sub-group was small, leading to limited power and wide Cis. Therefore, we could not determine whether associations were beneficial or harmful among specific sub-groups.”

Lines 240-262 – This section is not presented in a clear way. First, all of the studies considered are listed. Then, the list is reduced to just what is available for pooling, which is three potential studies. Then suddenly we are down to one study, one compound. The only findings of the other studies that are presented are those from the two considered for pooling that could not be pooled. At least some minimal comments need to be made on the overall findings related to 3PBA. And also, there should be more about the study they did pool with, what were the findings, how did the levels of exposure compare?

Thank you for raising these points. We have clarified on lines 258-285: “Three studies reported on at least one of the same pesticide biomarkers we assessed and on at least one child development domain we assessed, and thus were eligible for pooling [43,51,53]. One of these studies, which used a different child development assessment tool than we did (the Gesell Developmental Schedules), provided insufficient information to convert author-reported estimates to MDs or SMDs [43]. A second of these studies classified 3-PBA exposure as <LOD, medium, or high, and provided insufficient information to select a comparable exposure group [53]. No responses were received from the authors of these two studies to requests for data to enable pooling. The third study published sufficient information for pooling associations of 3-PBA, the only common pesticide biomarker between this study and our [51]. In that study, conducted in Southwest China, 3-PBA was assessed in urine samples from 357 women taken in each trimester of pregnancy (8-12 weeks’ gestation, 20-23 weeks’ gestation, and 32-35 weeks’ gestation). Child development was assessed using BSID-III at 1 year of age [51]. Exposure to 3-PBA during the first or third trimester was not associated with child cognitive, language, motor, socio-emotional, or adaptive development. However, higher exposure during the second trimester was associated with lower cognitive and language scores, but not with motor, socio-emotional or adaptive scores [51]. We selected the first trimester (8-12 weeks’ gestation) for pooling since urine samples in our study were collected primarily during the same window (mean gestational age 11.6±2.9 weeks in our study). In the China study, 3-PBA was detected in 85% of women (geometric mean 2.34 μg/g creatinine) [51]. We summarised estimates for the adjusted association between creatinine-adjusted pregnancy 3-PBA concentrations (μg/g creatinine) and child development composite scores. The pooled results found that pregnancy 3-PBA concentrations were not significantly associated with cognitive (MD 0.11 (95% CI -0.42, 0.64), p=0.69, I2=0.0%, p=0.43 (Figure 2)), language (MD -0.16 (-0.77, 0.45), p=0.61, I2=0.0%, p=0.47 (Figure 3)) or motor composite scores (MD -0.57 (-1.86, 0.72), p=0.39, I2=0.0%, p=0.16 (Figure 4)). The two studies that could not be pooled due to lack of comparable information or raw data found that pregnancy concentrations of 3-PBA were associated with poorer mental development at 24 months of age [53], but not at 36 months or with motor development at 24 or 36 months [53] or IQ at 7 years [44].”

Discussion:

Lines 296-298 – If part of the project was to do a meta analysis, shouldn’t the information on what was found in other OP studies be included in the results?

Findings from studies included in the systematic review but not in the meta-analysis are summarised on lines 299-309: “Other included studies found inconsistent associations between pregnancy concentrations of cis-DCCA and trans-DCCA and cognitive, language, and adaptive development at 1 year of age and IQ at 7 years of age [44,46,51]. Associations between pregnancy concentrations of diethylphosphate, dimethylphosphate, and dialkylphosphate and child motor and language development at 24 months of age were also inconsistent [45,47,48]. Pregnancy dimethylphosphate and diethylphosphate concentrations were inversely associated with infant behaviour [55], whereas pregnancy concentration of TCPY was not associated with child development at 3 or 7 years of age [43,44]. Lastly, pregnancy concentrations of propoxur and carbofuranphenol (metabolites of carbamate pesticides) were associated with lower motor development at 2 years of age [54] and lower adaptive and social development at 3 years of age [49], but not with IQ at 7 years of age [50].”

Lines 298-300 – instead of “showed”, “found no association”, evidence is not strong enough to “show” anything.

Revised throughout the manuscript, for example: “These findings were supported by our systematic review which found inconsistent associations between pregnancy exposure to organophosphates, pyrethroids, and carbamates and child development up to 7 years of age.” (lines 321-323)

Line 300-306 – Since no information on the modifications were presented in the results, this is really hard to interpret.

We have now provided more information on the results to facilitate interpretation: “In exploratory analyses to assess whether adjusted associations differed across child, maternal, and household characteristics, we found that child sex modified the associations between TCPY and language development, IMPY and motor development, and 3-PBA and motor development (Table in S5 Table). Maternal education modified the associations between 4-nitrophenol and motor development. Household income modified the associations between 3-PBA and language and motor development. Although interactions were significant (p<0.10), the number of observations in each sub-group was small, leading to limited power and wide Cis. Therefore, we could not determine whether associations were beneficial or harmful among specific sub-groups.” (lines 211-220)

Lines 351-360 – Before jumping straight to organic, it might make sense to work on programs to at least get usage down since they seem to be clearly overusing the chemical. It is likely much easier to at least come in line with Western usage than jump straight to organic. Also, you have a typo with your ().

Thank you for this suggestion. We have noted it on lines 390-393: “Given these challenges with organic farming, other approaches to reducing pesticide use/misuse in agriculture such as integrated pest management, government training and inspection of pesticide retailers, agricultural extension support to farmers, and bans on the production and import of highly hazardous pesticides should be explored.”

Lines 361-369 – I don’t recall the literature on pesticides and folate intake, but I recall there might be some findings showing less of an impact when moms take folate right from the beginning. If you are going to speculate on potential interventions, this is a well proven one from a public health standpoint and is very low cost. Could be good to mention in a sentence or two.

Thank you for this suggestion. We have noted it on lines 394-409: “In contexts where interventions to directly reduce pesticide exposure are infeasible or slow to initiate and generate change, protective approaches such as responsive stimulation and parenting interventions or maternal folate intake may help offset the negative effects of pesticide exposure on child development. Responsive stimulation and parenting interventions are effective in improving the caregiving environment and in turn child development [63], including those tested in Bangladesh [64–66]. However, evidence on if, and how, these types of interventions mediate the effects of pesticide exposure on child development is lacking. Further work can help to understand and test the potential synergistic or additive effects on child development of combining interventions to reduce pesticide exposure and stimulation interventions. With respect to maternal folate intake, evidence from high-income countries suggests that maternal folate intake during pregnancy can be protective against developmental neurotoxicity, particularly in children with developmental disorders [67,68]. For example, one study in the United States found that high folic acid intake (≥800 μg) early in pregnancy was associated with lower odds of autism spectrum disorder at age 2-5 years [67]. While more research is needed to confirm these findings, continuing to recommend that pregnant women consume iron-folic acid supplements [69] may have co-benefits by attenuating the adverse consequences of pesticide exposure on child development.”

Limitations: I think the following limitation needs to be mentioned somewhere. These woman were exposed to massive levels of organophosphate pesticides. It is not at all surprising that they could not find a relationship with a class of compounds that does not appear to be regularly used in Bangladesh (only 20% detects).

Thank for raising this point. This limitation is discussed on lines 339-349: “First, two important toxicology limitations should be noted. One is that all studies, including ours, assessed a few active ingredients. For example, the included studies conducted in China examined up to six active pesticide ingredients, while 239 active ingredients are currently approved in China [30]. Further, people are usually exposed to multiple active ingredients or complex mixtures, which were not examined in ours or other included studies. The second toxicology limitation is that most of the biomarkers assessed in the included studies were developed for use in HICs and do not match to frequently used active ingredients in LMICs. As evidenced in our sample, three of the pesticide biomarkers we quantified were not detected in most participants and one was not detected in any participants. From the four pesticide biomarkers we quantified, IMPY and 3-PBA were detected in fewer than 20% of women. Therefore, our results should be interpreted with caution.”

Conclusion: lines 374-377 – you can’t jump from 13 studies to “pooled analyses” when you only pooled with one study. See general comments

Thank you for this comment. We have revised to: “In analyses which pooled our study with another study, pregnancy concentrations of 3-PBA (a pyrethroid insecticide metabolite) were not associated with cognitive, language, or motor development in early childhood.” (lines 416-418).

---

## [Editor Report · Decision Letter 1]

31 May 2023

Pregnancy pesticide exposure and child development in low- and middle-income countries: a prospective analysis of a birth cohort in rural Bangladesh and meta-analysis

PONE-D-23-02593R1

Dear Dr. Bliznashka,

We’re pleased to inform you that your manuscript has been judged scientifically suitable for publication and will be formally accepted for publication once it meets all outstanding technical requirements.

Kind regards,

Iman Al-Saleh

Academic Editor

PLOS ONE

Additional Editor Comments (optional):

The authors have adequately addressed the comments raised by the reviewers in the revised version of the manuscript. Therefore, there are no further comments. Thank you.
---

## [Editor Report · Acceptance letter]

2 Jun 2023

PONE-D-23-02593R1 

Pregnancy pesticide exposure and child development in low- and middle-income countries: a prospective analysis of a birth cohort in rural Bangladesh and meta-analysis 

Dear Dr. Bliznashka:

I'm pleased to inform you that your manuscript has been deemed suitable for publication in PLOS ONE. Congratulations! Your manuscript is now with our production department. 

Kind regards, 

on behalf of

Dr. Iman Al-Saleh 

Academic Editor

PLOS ONE